# On-Match Impact and Outcomes of Scoring First in Professional European Female Football

**DOI:** 10.3390/ijerph182212009

**Published:** 2021-11-16

**Authors:** Patricia Sánchez-Murillo, Antonio Antúnez, Daniel Rojas-Valverde, Sergio J. Ibáñez

**Affiliations:** 1Research Group in Optimization of Training and Performance Sports, Faculty of Sport Sciences, University of Extremadura, 10005 Cáceres, Spain; psanchezmy@alumnos.unex.es (P.S.-M.); antunez@unex.es (A.A.); 2Centro de Investigación y Diagnóstico en Salud y Deporte (CIDISAD), Escuela Ciencias del Movimiento Humano y Calidad de Vida (CIEMHCAVI), Universidad Nacional, 86-3000 Heredia, Costa Rica; 3Clínica de Lesiones Deportivas (Rehab&Readapt), Escuela Ciencias del Movimiento Humano y Calidad de Vida (CIEMHCAVI), Universidad Nacional, 86-3000 Heredia, Costa Rica

**Keywords:** women, football, final score, winning, match result, situational variables

## Abstract

Background: Scoring first seems to be a determinant in professional football playing; several factors could influence the development of the match and the outcome. This study aimed to identify which factors could influence scoring first and impact match outcomes in professional European female football. Methods: There were 504 official matches held on 74 match days during the 2018–2019 professional female European football seasons (*Primera Iberdrola*, *D1 Féminine*, and *Frauen-Bundesliga*), analysed using a notational and inferential assessment. Results: There was a direct positive relationship (*p* < 0.05) between scoring first and winning the match; 75.9% of the winning teams scored first. Moreover, those teams that usually scored first had a better final league classification (*p* < 0.05). These relationships were not influenced by home or away conditions. Conclusions: Scoring first is a determinant in the outcomes of professional European female football matches. Physical and tactical training and programming should focus on those variables, leading female teams to score first.

## 1. Introduction

Football is well-known as one of the most played and popular sports worldwide. This sport attracts millions of fans, and the economic and social interest in related events, tournaments, and matches continues to grow. Curiously, despite its attractiveness, football match outcomes are often determined by relatively few critical actions, leading to small final scores. Usually, football matches have an average of 2.7 goals per game [1]. This reduced number of goals, which determines the final match result is the primary rationale of a study on the influence of scoring first in professional football [2].

Several studies analyse the impact of scoring first in male football [3,4,5]; but little studies focused on female teams [3]. All of the scientific evidence suggested that scoring first in female and male football is critical; this advantage increases the probability of winning the matches. In male football, 65–75% of the matches are won by the first-scoring teams, whereas female football lacks the evidence to summarise the real impact and benefits of scoring first.

Some factors may be determinant to scoring first, such as the home advantage, team league classification, and if the goal was scored in the first or second half of the match. The home advantage is understood as the relative advantage of being the match host. Previous studies in male football suggested that, when playing at home, the home team has winning odds of 74% [4]. Contrarily, if the visiting team scores first, the winning odds are 50% [5] to 63% [4]. These odds also seem to apply to women’s football [6]; in an analysis of the top Spanish league, those female teams that scored first in the match won in more than 80 to 90% of cases, depending on the team’s classification (top three vs. top ten) [7]. Additionally, in top European male football leagues, if the home team scores first, the winning odds are 62% [5].

The scoring-first effect on match outcome was studied in other sports, such as baseball and hockey. Scoring first (66.3%) and being the home team (61.7%) are determinants of the final score in baseball [8]. In hockey, the psychological momentum of scoring first causes a marked increase in the likelihood of winning the matches [9].

The abovementioned studies highlighted the second determinant of scoring first, proposing that the best-classified teams usually scored first during female matches [7]. This was also the cases in male football, where the best-ranked teams usually scored first and, as expected, won their matches [10]. Some evidence in male football suggested that higher-budget teams are more likely to win (14%) [11]. Additionally, the competitive balance, understood as the balance in the sport capabilities of teams, could influence the match outcome [12]. Additionally, other team characteristics, such as overall quality and overall ranking, are determinants in team sports [13,14].

Additionally, analyses of the top European championships (UEFA Champions League and European League) show that the first goal scored between 16 and 45 min of match time is more determinant in the outcome than those scored during the first 15 min of the match [15]. This evidence was confirmed by recent studies, suggesting that the team that scored its first goal during the final minutes of a game is usually the winner [1].

All of these situational factors could define the outcome of a match, and they are usually explained by a series of tactical and psychological reasons. Tactical passivity, decreased motivation and confidence, and reduced crew tactical structure and cohesion are common issues for losing teams after conceding a goal [16]. Additionally, with regard to the basic physiological differences [17], female football, as opposed to male football, has particular characteristics that could influence the match outcome and first goal, as budget disparities (competitive balance), quality, and technical and tactical skills differ between teams. To better understand female professional players’ psychological, tactical, and physical behaviour and acknowledge the lack of studies on women’s football, this study aimed to identify which factors could influence scoring first and how they could impact match outcomes in professional European female football.

## 2. Materials and Methods

### 2.1. Sample

This study was defined as observational since the authors did not influence the natural behaviour of the matches, using an ex post facto analysis [18]. All official matches were recorded and explored using a systematic notational analysis. 

A total of 504 official matches held on 74 match days during the 2018–2019 top female European football seasons (*Primera Iberdrola*, *D1 Féminine*, and *Frauen-Bundesliga*) were assessed. The distribution of the matches by league was as follows: Primera Iberdrola = 240, D1 Féminine = 132 and Frauen-Bundesliga = 132. The French and German leagues included 12 teams and a total of 22 match days, while the Spanish league included 16 teams and a total of 30 match days. Consequently, a total of 1008 cases were analysed. The format of all leagues defined the champion as the team with the most points (best-ranked) following both home and away matches.

All of the data analysed were extracted from a digital database, accessible to the public from the leagues’ official websites (e.g., www.laliga.es). This task was performed by two experts and was contrasted. If there were inconsistencies, both experts agreed after a consensus and final review of the databases.

### 2.2. Variables

Based on the previous literature, the selected variables were chosen as independent: first-scorer team (first scorer vs. the second scorer,) and the following as dependent variables: match time when the first goal was scored (e.g., 0–15 min, 16–30 min, 30–45 min, 45–60 min, 60–75 min, 75–90 min, 90+ min), league (*Primera Iberdrola* vs. *D1 Féminine* vs. *Frauen-Bundesliga*), local conditions (home vs. away). Other quantitative variables were used, including final ranking (top 1–4 vs. positions 5–12 and 13–16 for Primera Iberdrola; 5–8 and 9–12 for Frauen-Bundesliga and D1 Féminine), number of goals scored (0–11 per team), and number of yellow (0–6 per team) and red cards (0–1 per team).

### 2.3. Statistical Analysis

The observational data collection was performed using a data sheet (Excel, Microsoft Office 365, Mountain View, CA, United States). Analyses were made using the Statistical Package for Social Sciences (SPSS v.21.0, Chicago, IL, USA).

The normality of the data was explored by Kolmogorov–Smirnov test. Categorical variables were treated as non-parametrical data. The data were presented using descriptive analysis and frequency distribution for qualitative variables; for quantitative variables, the data were presented as mean, minimum, maximum and typic deviations [19]. Inferential analyses were used to explore the potential influence of independent variables on the dependent variable. Association between variables was explored using chi-squared (*X*^2^) and Cramer’s V (*V*). The relationship between variables using *X*^2^ was understood as *p* < 0.05. Cramer´s V was interpreted using previous criteria [20], as follows: trivial (<0.10), small (0.10–0.29), moderate (0.30–0.49) and large (>0.50). 

The relationship level between variables was established using the *adjusted standardised residuals (ASR)* and different contingency tables (Field, 2009). Those residuals greater than 1.96 confirmed the association between variables, the interpretation made based on previous criteria [21].

Differences in goals, yellow cards, and red cards between first and non-first scorers were explored using a one-way analysis of variance (*F* value) (1 × 3).

## 3. Results

The descriptive results of match outcome, local conditions and the match time when the first goal was scored are presented by league (see Table 1). In the Primera Iberdrola League, the winning teams scored first in 77.3% of cases (*X*^2^ = 393.5, *p* < 0.01; *V* = 0.6 *(large)*, *p* = 0 < 0.01, *ASR* = 15.2), D1 Féminine teams in 94.0% of cases (*X*^2^ = 192.8, *p* < 0.01; *V* = 0.6 *(large)*, *p* < 0.01, *ASR* = 10.9) and Frauen-Bundesliga in 95.0% of the matches (*X*^2^ = 174.0, *p* < 0.01; *V* = 0.6 *(large)*, *p*< 0.01, *ASR* = 11.1). 

Moreover, must of those teams that scored first did so in the first 15 min (36.9% for Primera Iberdrola, 36% for D1 Féminine and 43.8% for Frauen-Bundesliga). Additionally, the home teams and first scorers won in 52.9%, 55.2% and 53.9% of cases, respectively. There was no statistical evidence suggesting a clear probability of winning when scoring at a specific match time for any league (Primera Iberdrola: *X*^2^ = 1.5, *p* = 0.4; *V* = 0.1 *(small)*, *p* = 0.5, *ASR* = 1.2; D1 Féminine: *X*^2^ = 12.7, *p* = 0.3; *V* = 0.1 *(small)*, *p* = 0.3, *ASR* = 1.6; Frauen-Bundesliga: *X*^2^ = 1.6, *p* = 0.5; *V* = 0.1 *(small)*, *p* = 0.5, *ASR* = 1.2).

In Table 2, the descriptive data are shown relative to the number of goals and yellow /red cards, considering if the team scored first or not. First scorers scored a mean of 2.7 goals per match, significantly higher than non-first scorers. There were statistical differences in goals where first scorers scored more goals (*F* = 239.6, *p* < 0.01). No differences were found in number of yellow (*F* = 1.4, *p* = 0.2) or red (*F* = 0.1, *p* = 0.9) cards.

Finally, the better-ranked teams usually scored first in all leagues. The association between variables for the best-ranked teams was as follows: (Primera Iberdrola: *X*^2^ = 43.5, *p* < 0.01; *V* = 0.2 *(small)*, *p* = 0 < 0.01, *ASR* = 5.9; D1 Féminine: *X*^2^ = 30.3, *p* < 0.01; *V* = 0.2 *(small)*, *p* < 0.01, *ASR* = 5.1; Frauen-Bundesliga: *X*^2^ = 43.9, *p* < 0.01; *V* = 0.3 (*moderate)*, *p* < 0.01, *ASR* = 5.3).

## 4. Discussion

This study aimed to identify which factors could influence scoring first and match outcomes in professional European female football. The results of the analyses suggested that scoring first in female professional football was critical to winning. Female football teams in top European leagues who scored first won in 77–95% of the matches and were better ranked, significantly different results from non-first scorers. More than a third of the first scorers’ teams scored in the first 15 min (36–43% of cases) but with no statistical differences compared to other match time points. First scorers scored more goals, but no differences were found in the number of yellow or red cards. Local status also did not influence the final match outcome.

The results of the study confirmed the critical role of scoring first in female professional football. Scoring the first goal may create an advantage, psychologically, tactically, and physically. The evidence suggests that when male football teams are winning, the team usually creates a positive psychological momentum and mindset that makes winning more probable [9]. When scoring first, the conceding team tend to have a higher ball possession than when drawing or wining [22,23]. Additionally, losing teams tend to make more mistakes (e.g., ball interceptions by a rival, fewer clearances) [24] and show more high-intensity actions [14,25], positioning the losing team in a technically and tactically disadvantageous position. Moreover, female losing teams make more tackles, lose the ball more often and accumulate more yellow and red cards than the winning teams [26]. 

Compared to other sports (50–65%) [8], in female football (77–95%) the odds of winning after scoring first are higher. Compared to male football (65–75%) [3,4,5], the probability of scoring first in female football is slightly higher. Additionally, these odds depend on the league and could be explained by the greater difference in the teams’ quality and the competitive imbalance of female football compared to male football. This could be explained by the higher heterogeneity in the quality of female football compared to male football, with professional and semi-professional female players competing in the same league.

Furthermore, the match time point in which the first goal is scored is also a variable that was studied [4]. Critical moments of the match were identified as essential and are in accordance with the results of this study. The first 15 min of both halves and the last 15 min of the match are critical periods regarding scoring first [27]. These critical moments in the female matches usually see an increase in match workload; depending on the game situation, this increase can be between 20 and 25% [28]. This increase in some periods of the match should be considered when addressing periodisation and strategies to achieve the first goal.

Finally, the evidence suggests that home advantage depends more on the quality of the home team and its rival than on a home effect per se [29]. Indeed, the results of this study indicate that being the first scorer is a determinant regardless of whether the scoring team is the home or the away team. It is also known that teams playing at home tend to show a higher ball possession than teams playing away [22]. Nonetheless, when playing against a stronger opponent, the opposing team must perform better [30], which may cause physical exhaustion and alterations to tactics. In this sense, the best quality team have a more stable pattern of play [13], improving their ability to perform consistent high-intensity actions (e.g., sprints, high-speed running, accelerations, changes of direction), which are essential for physical improvements in female football [31,32]. 

In female football, the home advantage effect was not as high as in male football. It seems that, in the European leagues, the home advantage in female football is reflected in winning odds of 51–59%; in male football the odds are almost 60% [6]. Some factors could explain these differences, such as the crowd effect (size, intensity and density) on players and referees (referee bias) and gender perceptions of territorial protection and competitive balance [33], usually greater in male football [6]. The aggressivity of the sport and intensity of the match also influence the home advantage. These plausible reasons are supported by other sports studies of the differences in home advantage by gender (e.g., water polo, handball) [34,35]. In Western European football, the evidence suggests a decrease in the home advantage, compared to that of recent decades, due to changes in rules, sport structure and diffusion since the 1980s, leading to a competitive balance [33,36].

Finally, recent studies proposed some tactical, technical and physical factors that may increase the likelihood of winning in female football and could influence scoring first. For example, one-quarter of goals are scored from crosses [37], and so free kicks should be made by a direct free kick or a direct shot on goal [38], pass accuracy should be increased, and a better performance in offensive and defensive duels should be demonstrated [39]. These are technical and tactically significant contributors to victory. Therefore, these actions and situations should be incorporated into training and improved upon to increase the odds of scoring the first goal and winning the match. Additionally, regarding tactics, the winning teams often intercept and recover the ball in more advanced regions of the field than the losing teams [40]; this could suggest the need for some deep pressure strategies during the match, resulting in a higher number of goal attempts. Additionally, some key indicators, such as the high-intensity actions of sprinting, running distances, high ball possession and optimal attacking organisation could influence the match outcome [41]. A home disadvantage is supported by a hypothesis underlying the pressure of winning in front of a supportive audience (expectation of winning). There is a diffusion of responsibility among team members in football compared to other sports, such as basketball or baseball [42]. In other sports, such as hockey and rugby, there is a hypothesis regarding the inhibition of anxiety, which reduces pressure on the home team due to high physical contact [42]. 

Due to the common playing dynamics and considering that football is a low-scoring sport, with an average of three or fewer goals [1], being the first scoring team seems determinant in female professional football leagues [27]. Scoring first has a strong positive effect that influences the match outcome. These findings can be fundamental for football coaches when developing strategic and tactical planning to enhance the performance of their players, with regard to the different situational variables that their teams may face during the matches.

### Limitations

The main limitation of this study is due to the insufficient number of published scientific articles, which analyse the situational and contextual variables that influence female professional football, specifically those focusing on official match analyses and those factors that impact the final match outcomes. Several studies explored the situational and conditional factors that could influence performance in male football. Based on the basic physiological, genetic, social and cultural differences between women and men [17], such information must be collected and analysed concerning performance and the factors that influence performance in the female sphere. This gender gap reveals the need to better explore female performance in professional football in future studies.

Additional limitations are based on the limited data availability of the different official web pages, with regard to the specification of players’ characteristics, weather conditions, and other situational information that could affect the interpretation of the data. While this study analysed the 2018–2019 tournaments of the Spanish, German and French female professional football leagues, recent data are not available or conclusive due to the season cancellations cause by the COVID-19 pandemic.

## 5. Conclusions

Scoring first determines the outcomes of official matches from top female European leagues and impacts the final number of goals in a match and the final ranking of the league team. Additionally, it is well known that scoring first provides an advantage during matches and could condition game dynamics, tactics, and programming during a tournament. The top-ranked teams usually have better preparation processes and resources available to develop physical, psychological and tactical skills; these advantages allow them to score the first goal in most cases.

### Practical Applications

Considering the impact of being the first scorer in a female football match, coaches may plan training sessions, bearing in mind that there are physical and psychological aspects to focus on that may boost team performance during the opening minutes of a match, and design tasks that may help achieve the first goal. Different offensive strategies may also help when taking into account physical conditioning, the quality of the opposing team, the available players and their abilities.

Additionally, some training strategies may be devised to overcome a situation where the team is not the first scorer, thus avoiding negativity and encouraging confidence in solving this issue. Moreover, those teams at the bottom of the table may focus their training towards achieving the first goal as a condition that may change the outcome of the matches.

The teams from minor divisions should be aware of these results and consider the critical opening minutes of a match as a determinant for winning the match, and thus the fundamental role of scoring first. Finally, coaches may analyse these tactical and physical situations where the team scores the first goal to understand how it was achieved to boost future performance, considering the variables that may help accomplish the first goal.

The internal and external variables mediating the scoring of the first goal or the offensive tactics that may lead to it should be monitored and considered when planning physical and conditioning training (e.g., high-intensity actions, accelerations, changes of direction).

Future studies could focus on the teams who score first, considering psychological, tactical and physical conditions. Additionally, studies could explore how female footballers, whose teams are losing during a match, can overturn the match outcome.

## Figures and Tables

**Table 1 ijerph-18-12009-t001:** Descriptive analysis of the result, time point when the first goal was scored, and local status by the league and first scorer.

League	First Scorer	Result			First Goal Time Point							Local Status	
		Won n (%)	Loss n (%)	Draw n (%)	No Goal (%)	0–15 min n (%)	16–30 min n (%)	31–45 min n (%)	45–60 min n (%)	61–75 min n (%)	75–90 min n (%)	Home n (%)	Visit n (%)
Primera Iberdrola	Yes	174.0 (77.3)	23.0 (10.2)	28.0 (12.4)	0.0 (0.0)	83.0 (36.9)	44.0 (19.6)	46.0 (20.4)	21.0 (9.3)	20.0 (8.9)	11.0 (4.9)	119.0 (52.9)	106.0 (47.1)
	No	23.0 (10.2)	174.0 (77.3)	28.0 (12.4)	225.0 (100.0)	0.0 (0.0)	0.0 (0.0)	0.0 (0.0)	0.0 (0.0)	0.0 (0.0)	0.0 (0.0)	106.0 (47.1)	119.0 (52.9)
	Draw	0.0 (0.0)	0.0 (0.0)	30.0 (100)	30.0 (100.0)	0.0 (0.0)	0.0 (0.0)	0.0 (0.0)	0.0 (0.0)	0.0 (0.0)	0.0 (0.0)	15.0 (50.0)	15.0 (50.0)
D1 Féminine	Yes	94.0 (75.4)	13 (10.4)	18.0 (14.4)	0.0 (0.0)	45.0 (36.0)	31.0 (24.8)	25.0 (20.0)	9.0 (7.2)	10.0 (8.0)	5.0 (4.0)	69.0 (55.2)	56.0 (44.8)
	No	13.0 (10.4)	94.0 (75.2)	18.0 (14.4)	125.0 (100.0)	0.0 (0.0)	0.0 (0.0)	0.0 (0.0)	0.0 (0.0)	0.0 (0.0)	0.0 (0.0)	56.0 (44.8)	69.0 (55.2)
	Draw	0.0 (0.0)	0.0 (0.0)	14.0 (100.0)	14.0 (100.0)	0.0 (0.0)	0.0 (0.0)	0.0 (0.0)	0.0 (0.0)	0.0 (0.0)	0.0 (0.0)	7.0 (50.0)	7.0 (50)
Frauen-Bundesliga	Yes	95.0 (74.2)	10.0 (7.8)	23.0 (18.0)	0.0 (0.0)	56.0 (43.8)	31.0 (24.2)	20.0 (15.6)	14.0 (10.9)	4.0 (3.1)	3.0 (2.3)	69.0 (53.9)	59.0 (46.1)
	No	10.0 (7.8)	95.0 (74.2)	23.0 (18.0)	128.0 (100.0)	0.0 (0.0)	0.0 (0.0)	0.0 (0.0)	0.0 (0.0)	0.0 (0.0)	0.0 (0.0)	59.0 (46.1)	69.0 (53.9)
	Draw	0.0 (0.0)	0.0 (0.0)	8.0 (100.0)	8.0 (100.0)	0.0 (0.0)	0.0 (0.0)	0.0 (0.0)	0.0 (0.0)	0.0 (0.0)	0.0 (0.0)	4.0 (50)	4.0 (50)

**Table 2 ijerph-18-12009-t002:** Descriptive data of final number of goals and red/yellow cards by the first scorer.

First Scorer	Variable	n Cases	Minimum	Maximum	Mean	Typical Deviation
Yes	Goals (n)	478	1.0	11.0	2.7	1.8
	Yellow cards (n)	478	0.0	6.0	1.1	1.1
	Red cards (n)	478	0.0	1.0	0.1	0.2
No	Goals (n)	478	0.0	10.0	0.8	1.1
	Yellow cards (n)	478	0.0	5.0	1.3	1.0
	Red cards (n)	478	0.0	1.0	0.1	0.20
No goal	Goals (n)	52	0.0	0.0	0.0	0.0
	Yellow cards (n)	52	0.0	4.0	1.2	1.0
	Red cards (n)	52	0.0	1.0	0.1	0.2

## Data Availability

Not applicable.

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
