# Peer review of "On-Match Impact and Outcomes of Scoring First in Professional European Female Football"

_ijerph, 2021, doi:10.3390/ijerph182212009_

Round 1

Reviewer 1 Report

I really appreciate the novelity of the work. There is a lack of studies confirming the real impact of the benefit of being the first scorer in female football.  The authors decided to fill the gap and did it very interesting with a total number of 504 official matches analyzed. 

I have no major comments, only minor - presented below.

How physical, tactical, psychological training and programming can affect to score first?

There are some typos: relation instead Relation (p. 3, l.102), Budesliga instead Bundesliga (Table 2) and so on. The Authors should read whole manusript very carefully and correct all typos.

The sentence: Top-ranked teams are usually more physically, psychologically, and tactically ; these advantages allow them to achieve the first goal in most cases. (p. 6, l. 211) is a little unclear in my opinion. I don't think that a team can be more physically, psychologically, and tactically, rather better prepared physically, psychologically, and tactically.

I believe that Table 1 is not necessary, its content can be described in text.

Author Response

Dear Editor and reviewers:

We have carefully considered all reviewers' recommendations for the paper (ijerph-1439346) entitled " On-match impact and outcomes of scoring first in professional European female football”. Please find enclosed our detailed answers to reviewers' queries. The authors declare that the manuscript is original and has not been considered for publication elsewhere. Additionally, the authors had approved the paper for release and agreed with its content.

Please find all corrections in red inside the manuscript.

Reviewer: 1
R1.1 I really appreciate the novelty of the work. There is a lack of studies confirming the real impact of the benefit of being the first scorer in female football.  The authors decided to fill the gap and did it very interesting with a total number of 504 official matches analyzed. 

I have no major comments, only minor - presented below.

R/We really appreciate the reviewer for it time and dedication to review this manuscript. We have reviewed and considered all his/her recommendations and corrections.

How physical, tactical, psychological training and programming can affect to score first?

R/We have included this clarification in Discussion section.

R1.2. There are some typos: relation instead Relation (p. 3, l.102), Budesliga instead Bundesliga (Table 2) and so on. The Authors should read whole manusript very carefully and correct all typos.

R/Thank you for pointing out these issues, we have review and corrected all typos.

R1.3.The sentence: Top-ranked teams are usually more physically, psychologically, and tactically ; these advantages allow them to achieve the first goal in most cases. (p. 6, l. 211) is a little unclear in my opinion. I don't think that a team can be more physically, psychologically, and tactically, rather better prepared physically, psychologically, and tactically.

R/We have corrected this issue, we agree with the reviewer.

R1.4I believe that Table 1 is not necessary, its content can be described in text.

R/As suggested the table 1 was deleted.

Reviewer: 2

R2.1.The study assessed the effect of scoring first on European female football leagues (Spain, France, and Germany) across one competitive season. It is a topic well analysed in the scientific literature in different team sports. Whilst the study undoubtedly has merit, it is necessary to revise. Moreover, I would recommend that a native co-author review the manuscript. The study could be potentially interesting but should be improved in the following aspects:

R/We really appreciate the reviewer for it time and dedication to review this manuscript. We have reviewed and considered all his/her recommendations and corrections.

INTRODUCTION

R2.2. Despite a good review of the problem, I consider that it would be appropriate to introduce more information from the studies provided (%, OR, etc.).

R/As suggested we have add some information regarding of the winning odds considering some situational variables.

R2.3.L35-40: Why is important to score the first goal? It is a well-studied phenomenon, so consider explaining the different reasons that previous authors have reported.

Courneya, K. S. (1990). Importance of game location and scoring first in college baseball. Perceptual and Motor Skills, 71(2), 624–626. https://doi.org/10.2466/pms.1990.71.2.624

Jones, B. M. (2009). Scoring first and home advantage in the NHL. International Journal of Performance Analysis in Sport, 9(3), 320–331. https://doi.org/10.1080/24748668.2009.1186848 9

Moreover, I think will enrich this paragraph if you introduce some examples of other team sports.

R/We have add the evidence suggested by the reviewers in the introduction section.

R2.4.L41-49: I would suggest expanding and delving deeper into the literature, I consider that this section is not sufficiently justified with the evidence shown. I consider it would be necessary to mention the different situational variables (HA, teams’ level, quality of opposition, etc.) and how the interaction between them could affect match outcomes.

R/ This section have been supported with more and more in depth evidence.

R2.5.Line 43: I would suggest defining Home Advantage effect, before introducing the influence on teams’ performance.

R/Home advantage was defined in this section previous to explain it impact on performance.

R2.6.Line 59-66: Why should be interested search the ‘scoring first effect’ from a gender approach? Please answer around this place.

R/This rationale was explained in the text, last paragraph introduction.

MATERIAL AND METHODS

R2.7.Did you check the sample normality? The number of matches is pretty different from leagues.

R/Thank you for the opportunity to clarify, this was considered at the 2.3. section.

DISCUSSION

R2.8.I think authors should compare more their findings with current literature. Especially with studies that have analysed the influence of scoring first in different team sports. On the other hand, authors should compare their results with male leagues.

R/Thank you for highlight this recommendation we have made changes accordingly.

R2.9.Are the values similar? If not, explain the reasons plausible reasons, please. For example, genetic aspects, cultural aspects, etc. Please, you will need fresh and newest references.

R/These aspects were approached in the discussion and limitation sections.

R2.10.Line 171-180: Please, explain why game location does not affect scoring the first goal in female football. I found this finding very important taking into consideration the current literature about this topic. Try to explain the differences between the male counterparts.

R/This was explained more in depth in the discussion section. Large amount of evidence was added to support these results.

R2.11.Moreover, it could be interesting to include more examples of poor performance in team sports when playing at home compared to playing abroad. I suggest including some of the following references:

Arboix-Alió, J., Trabal, G., Valente-Dos-Santos, J., Aguilera-Castells, J., Fort-Vanmeerhaeghe, A., & Buscà, B. (2021). The influence of contextual variables on individual set-pieces in elite rink hockey. International Journal of Performance Analysis in Sport, 21(3), 336–347. https://doi.org/10.1080/24748668.2021.1890525

Loignon, A., Gayton, W., Brown, M., Steinroeder, W., & Johnson, C. (2007) Home disadvantage in professional ice hockey. Perceptual and motor skills, 104(3c), 1262–1264

R/ This was explained more in depth in the discussion section. Large amount of evidence was added to support these results.

R2.12 How does teams’ quality influences the effect of scoring first? I think it should be mentioned the concept of competitive balance in female leagues since it is very common (take as example Primera Iberdrola) that exists an evident level of bias, caused probably by the different budgets of teams competing in the same division. This issue causes more heterogeneity of level than in male leagues, with professional and semi-professional athletes in the same competitions.

R/Please see corrections in both introduction and discussion section were authors have analyze this factor more in depth.

Reviewer: 3

R3.1The article has merit. However, minor revisions must be done before acceptance [see Guide for Authors].

R/We really appreciate the reviewer for it time and dedication to review this manuscript. We have reviewed and considered all his/her recommendations and corrections.

R3.2.Minor spell check required. Furthermore, in this paper is necessary to improve some aspects.

R3.3.In section 2, Materials and Methods, it’s necessary to improve the write this section, because the Statistical analysis isn´t clear and does not contain the mathematical notations correctly. In subsection 2.1 without any information about the characterization of the sample.

R/Please see section to analyze new approach.

R3.4.In section 3, Results, it is necessary to improve the writing of this section because the mathematical notations do not help to understand. Suggest putting tables in the appendix.

R/The results were better explained.

R3.5.In section 5, Conclusions, it is necessary to improve the writing of the main contribution of this study.

R/This sections was better explained in the new version of the manuscript.

R3.6.In section 6, Practical Applications, it is necessary to improve the writing of the main Practical Applications of this study.

R/This sections was better explained in the new version of the manuscript.

R3.7.References should be described as follows in “Instructions for Authors” (https://www.mdpi.com/journal/ijerph/instructions).

R/Reference were corrected following instructions for authors guideliness.

Reviewer: 4

R3.4.The work presented provides extensive bibliographic support, although there is not much bibliography on the matter about women's team sport, if there is an extensive bibliography on the variable, write down first. The methodological approach is appropriate, as is the statistical treatment. Highlight the care in the discussion section in which the authors have provided abundant considerations about their research and previous studies. The conclusions are clear and precise and it is valued that they are applicable to the field of practical training.Thank the authors for their interest in research in women's sports. Without a doubt, the conclusions of this study will help coaches to improve their knowledge of the game and prepare their athletes for professional competitions.

R/We really appreciate the reviewer´s commentary, we agree that women´s team sport needs more studies and lack of evidence. We support alternative sports, gender equality and other populations in disadvantage.

Reviewer 2 Report

Basic reporting

The study assessed the effect of scoring first on European female football leagues (Spain, France, and Germany) across one competitive season. It is a topic well analysed in the scientific literature in different team sports. Whilst the study undoubtedly has merit, it is necessary to revise. Moreover, I would recommend that a native co-author review the manuscript. The study could be potentially interesting but should be improved in the following aspects:

INTRODUCTION

Despite a good review of the problem, I consider that it would be appropriate to introduce more information from the studies provided (%, OR, etc.).

L35-40: Why is important to score the first goal? It is a well-studied phenomenon, so consider explaining the different reasons that previous authors have reported.

  • Courneya, K. S. (1990). Importance of game location and scoring first in college baseball. Perceptual and Motor Skills, 71(2), 624–626. https://doi.org/10.2466/pms.1990.71.2.624
  • Jones, B. M. (2009). Scoring first and home advantage in the NHL. International Journal of Performance Analysis in Sport, 9(3), 320–331. https://doi.org/10.1080/24748668.2009.1186848 9

Moreover, I think will enrich this paragraph if you introduce some examples of other team sports.

L41-49: I would suggest expanding and delving deeper into the literature, I consider that this section is not sufficiently justified with the evidence shown. I consider it would be necessary to mention the different situational variables (HA, teams’ level, quality of opposition, etc.) and how the interaction between them could affect match outcomes.

Line 43: I would suggest defining Home Advantage effect, before introducing the influence on teams’ performance.

Line 59-66: Why should be interested search the ‘scoring first effect’ from a gender approach? Please answer around this place.

MATERIAL AND METHODS

  • Did you check the sample normality? The number of matches is pretty different from leagues.

DISCUSSION

  • I think authors should compare more their findings with current literature. Especially with studies that have analysed the influence of scoring first in different team sports. On the other hand, authors should compare their results with male leagues.
  • Are the values similar? If not, explain the reasons plausible reasons, please. For example, genetic aspects, cultural aspects, etc. Please, you will need fresh and newest references.
  • Line 171-180: Please, explain why game location does not affect scoring the first goal in female football. I found this finding very important taking into consideration the current literature about this topic. Try to explain the differences between the male counterparts.
  • Moreover, it could be interesting to include more examples of poor performance in team sports when playing at home compared to playing abroad. I suggest including some of the following references:
    • Arboix-Alió, J., Trabal, G., Valente-Dos-Santos, J., Aguilera-Castells, J., Fort-Vanmeerhaeghe, A., & Buscà, B. (2021). The influence of contextual variables on individual set-pieces in elite rink hockey. International Journal of Performance Analysis in Sport, 21(3), 336–347. https://doi.org/10.1080/24748668.2021.1890525
    • Loignon, A., Gayton, W., Brown, M., Steinroeder, W., & Johnson, C. (2007) Home disadvantage in professional ice hockey. Perceptual and motor skills, 104(3c), 1262–1264
  • How does teams’ quality influences the effect of scoring first? I think it should be mentioned the concept of competitive balance in female leagues since it is very common (take as example Primera Iberdrola) that exists an evident level of bias, caused probably by the different budgets of teams competing in the same division. This issue causes more heterogeneity of level than in male leagues, with professional and semi-professional athletes in the same competitions.

Author Response

(The authors gave the same response as above.)

Reviewer 3 Report

The article has merit. However, minor revisions must be done before acceptance [see Guide for Authors].

Minor spell check required. Furthermore, in this paper is necessary to improve some aspects.

1. In section 2, Materials and Methods, it’s necessary to improve the write this section, because the Statistical analysis isn´t clear and does not contain the mathematical notations correctly. In subsection 2.1 without any information about the characterization of the sample.

2. In section 3, Results, it is necessary to improve the writing of this section because the mathematical notations do not help to understand. Suggest putting tables in the appendix.

3. In section 5, Conclusions, it is necessary to improve the writing of the main contribution of this study.

4. In section 6, Practical Applications, it is necessary to improve the writing of the main Practical Applications of this study.

5. References should be described as follows in “Instructions for Authors” (https://www.mdpi.com/journal/ijerph/instructions).

Author Response

(The authors gave the same response as above.)

Reviewer 4 Report

The work presented provides extensive bibliographic support, although there is not much bibliography on the matter about women's team sport, if there is an extensive bibliography on the variable, write down first. The methodological approach is appropriate, as is the statistical treatment. Highlight the care in the discussion section in which the authors have provided abundant considerations about their research and previous studies. The conclusions are clear and precise and it is valued that they are applicable to the field of practical training. Thank the authors for their interest in research in women's sports. Without a doubt, the conclusions of this study will help coaches to improve their knowledge of the game and prepare their athletes for professional competitions.

Author Response

(The authors gave the same response as above.)
